# TopoFR: A Closer Look at Topology Alignment on Face Recognition

**Jun Dan**[*1,2], **Yang Liu**[*2,3], **Jiankang Deng**[4], **Haoyu Xie**[2,5], **Siyuan Li**[2],
**Baigui Sun**[†2,5], **Shan Luo**[3]

[1]Zhejiang University [2]FaceChain Community [3]King's College London
[4]Imperial College London [5]Alibaba Group

danjun@zju.edu.cn, {yang.15.liu, shan.luo}@kcl.ac.uk, j.deng16@imperial.ac.uk
xiehaoyu.xhy@alibaba-inc.com, sunbaigui85@gmail.com

## Abstract

The field of face recognition (FR) has undergone significant advancements with the rise of deep learning. Recently, the success of unsupervised learning and graph neural networks has demonstrated the effectiveness of data structure information. Considering that the FR task can leverage large-scale training data, which intrinsically contains significant structure information, we aim to investigate how to encode such critical structure information into the latent space. As revealed from our observations, directly aligning the structure information between the input and latent spaces inevitably suffers from an overfitting problem, leading to a structure collapse phenomenon in the latent space. To address this problem, we propose TopoFR, a novel FR model that leverages a topological structure alignment strategy called PTSA and a hard sample mining strategy named SDE. Concretely, PTSA uses persistent homology to align the topological structures of the input and latent spaces, effectively preserving the structure information and improving the generalization performance of FR model. To mitigate the impact of hard samples on the latent space structure, SDE accurately identifies hard samples by automatically computing structure damage score (SDS) for each sample, and directs the model to prioritize optimizing these samples. Experimental results on popular face benchmarks demonstrate the superiority of our TopoFR over the state-of-the-art methods. Code and models are available at: https://github.com/modelscope/facechain/tree/main/face_module/TopoFR.

## 1 Introduction

Face recognition (FR) is a critical biometric authentication technique that is widely applied in various applications. In recent years, convolutional neural networks (CNNs) have achieved remarkable success in FR task, thanks to their powerful ability to autonomously extract face features from images. Existing studies on FR primarily focuses on constructing more discriminative face features by developing margin-based loss functions [1, 2, 3, 4, 5] and powerful network architectures [6, 7, 8, 9]. Recently, the success of unsupervised learning [10, 11, 12, 13, 14] and graph neural networks [15, 16, 17] has demonstrated the importance of data structure information in improving model generalization. However, to the best of our knowledge, how to effectively mine the potential structure information in large-scale face data has not investigated. Thus, in this paper, we extend our interests on building a cutting-edge FR framework through exploiting such powerful and substantial structure information.

First, we use Persistent Homology (**PH**) [20, 21], a mathematical tool used in topological data analysis [22] to capture the underlying topological structure of complex point clouds, to investigate

---
*Equal Contribution, † Corresponding Author.

38th Conference on Neural Information Processing Systems (NeurIPS 2024).

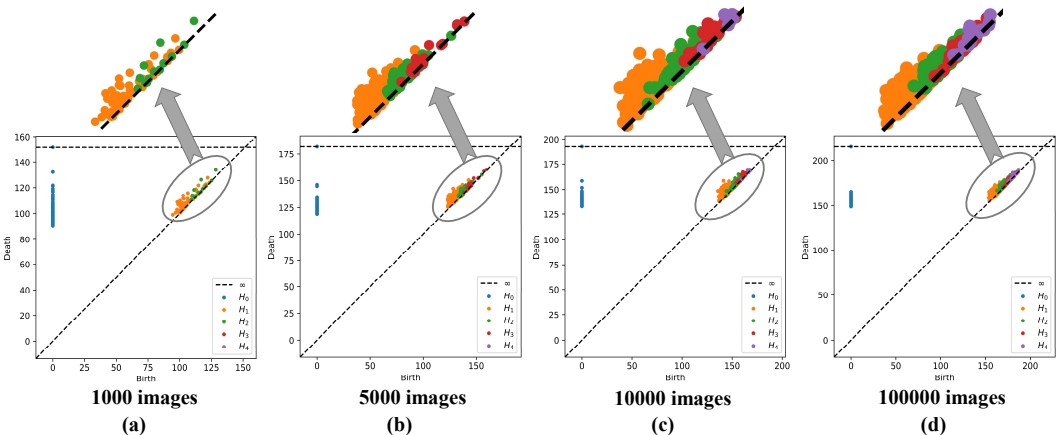

Figure 1: We sample 1000 (a), 5000 (b), 10000 (c) and 100000 (d) face images from the MS1MV2 dataset respectively, and compute their persistence diagrams using PH, where $H_j$ represents the $j$-th dimension homology. Persistence diagram [18] is a mathematical tool to describe the topological structure of space, where the $j$-th dimension homology $H_j$ in persistence diagram represents the $j$-th dimension hole in space. In topology theory, if the number of high-dimensional holes in the space is more, then the underlying topological structure of the space is more complex [19]. As shown in Figure 1(a)-1(d), as the amount of face data increases, the persistence diagram of the input space contains more and more high-dimensional holes (*e.g.*, $H_3$ and $H_4$). Therefore, this phenomenon demonstrates a growing complexity in the topological structure of the input space.

the evolution trend of structure information in existing FR framework and illustrate 3 interesting observations: **(1)** as the amount of data increases, the topological structure of the input space becomes more and more complex, as verified in Figures 1a-1d; **(2)** as the amount of data increases, the topological structure discrepancy between the input space and the latent space becomes increasingly larger, as verified in Figure 2a; **(3)** The results in Figure 2b demonstrate that as the depth of the network increases, the topological structure discrepancy becomes progressively smaller. This finding also provides an explanation for why models with more complex structure achieve higher FR accuracy. Based on the above observations, we can infer that in FR tasks with large-scale datasets, the structure of face data will be severely destroyed during training, which limits the generalization ability of FR models in practical application scenarios. To this end, we propose to improve the generalization performance of FR models by preserving the structure information.

However, we experimentally find that directly using PH to align the topological structures of the input space and the latent space may cause the model to suffer from **structure collapse phenomenon**. Concretely, under this experimental setting, we have 2 following quantitative results: **(1)** As shown in Figure 2c, the topological structure discrepancy drops to 0 dramatically during early training. **(2)** As depicted in Figure 2d, when evaluating on the IJB-C benchmark [23], there exists a significant structure information gap between the input space and the latent space. These typical overfitting phenomena indicate the latent space fails to preserve the structure information of input space accurately.

To remedy this issue, we propose a superior FR model named **TopoFR** that leverages a Perturbation-guided Topological Structure Alignment (**PTSA**) strategy to adequately preserve the topological structure information of the input space in corresponding latent face features. PTSA first employs a random structure perturbation (**RSP**) mechanism perturb the latent space and effectively increase its structure diversity. Then PTSA utilizes an invariant structure alignment (**ISA**) mechanism to align the topological structures of the original input space and the perturbed latent space, resulting in face features with stronger generalization ability

Moreover, in practical FR scenarios, the training dataset typically includes some low-quality face samples (*i.e.*, hard samples) that are prone to being encoded into abnormal positions close to the decision boundary in the latent space [24, 6, 25, 26], significantly destroying the topological structure of the latent space and affecting the alignment of structure. To address this issue, we propose a novel hard sample mining strategy named Structure Damage Estimation (**SDE**). SDE adaptively assigns structure damage score (**SDS**) to each sample based on its prediction uncertainty and prediction

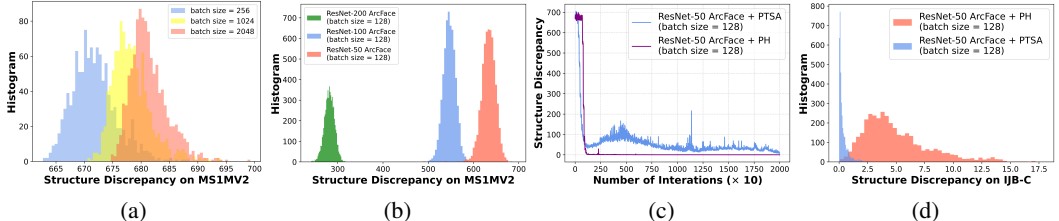

Figure 2: **(a):** We investigate the relationship between the amount of data and the topological structure discrepancy by employing ResNet-50 ArcFace model [1] to perform inferences on MS1MV2 training set. Inferences are conducted for 1000 iterations with batch sizes of 256, 1024, and 2048, respectively. Histograms are used to approximate these discrepancy distributions. **(b):** We investigate the relationship between the network depth and the topological structure discrepancy by performing inference on MS1MV2 training set (batch size=128) using ArcFace models with different backbones. **(c):** We investigate the trend of topological structure discrepancy during training (batch size=128) and found that **i)** directly using PH to align the topological structures will cause the discrepancy to drops to 0 dramatically; **ii)** whereas using our PTSA strategy promotes a smooth convergence of structure discrepancy. **(d):** Aligning the topological structures directly using PH will lead to significant discrepancy when evaluating on IJB-C benchmark. Our PTSA strategy effectively mitigates this overfitting issue, resulting in smaller structure discrepancy during evaluation.

probability. By prioritizing the optimization of hard samples with significant structure damage, SDE can gradually guide these samples back to their reasonable positions, thereby improving the generalization ability of FR model.

In summary, the main contributions are listed as follows:

**1)** To the best of knowledge, we are the first to explore the topological structure alignment in FR task. We propose a novel topological structure alignment strategy called PTSA to effectively align the structures of the original input space and the perturbed latent space.

**2)** A novel hard sample mining strategy named SDE is introduced to mitigate the adverse impact of hard samples on the latent space structure.

**3)** Experimental results show that the proposed method outperforms SOTA methods on various face benchmarks. Notably, our TopoFR has secured **the second place** in the ICCV21 MFR-Ongoing challenge [27] until the submission of this work (May 22 '24, academic track): `http://iccv21-mfr.com/#/leaderboard/academic`, indicating the robustness and generalization of our method.

## 2 Related Works

**Face Recognition (FR).** Convolutional Neural Networks (CNNs) [28, 29] have achieved remarkable advancements in tasks related to facial recognition [30, 31, 32, 1, 33, 34]. Notably, the extraction of robust deep facial embeddings has raised considerable interest within the research community. Among them, CNNs framework are representative methods, using two primary methods: metric learning-based and margin-based softmax approaches. The former utilizes loss functions like Triplet loss [7], Tuplet loss [35], and Center loss [36] to learn discriminative face features, while the latter aims to incorporate margin penalty into the softmax loss framework, including methods such as ArcFace [1], CosFace [2], AM-softmax [37], and SphereFace [38]. Recent studies have explored various techniques, including adaptive parameters [3, 5], mining [4, 39, 40], learning acceleration [41, 42, 43], vision transformer architecture [9, 8], and data uncertainty [25, 6, 26] to further enhance models' performance on large-scale datasets.

**Persistent Homology (PH).** Over the past decade, PH has shown significant advantages in multiple various such as signal processing[44], video analysis [45, 46], neuroscience [47, 48], disease diagnosis [49] and evaluation of embedding strategies [50, 51]. In the field of machine learning, some studies [52, 53, 54] have shown that integrating topological representations into network can enhance model's recognition/segmentation performance. [55] proposes a topology distance for the evaluation of GANs.

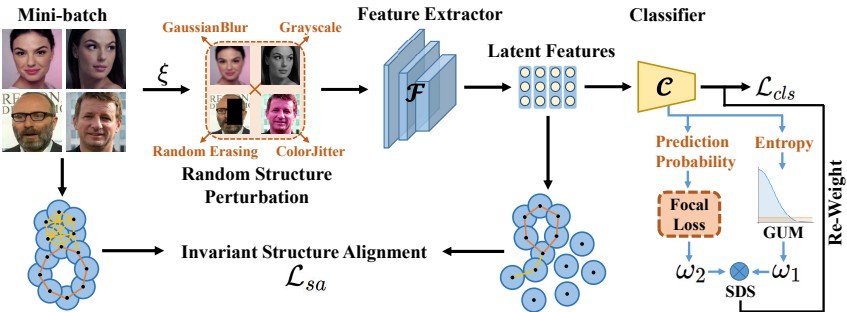

Figure 3: Global overview of our proposed TopoFR. $\bigotimes$ represents the multiplication operation. $\xi$ denotes the probability of applying RSP to each training sample.

## 3 Background: Persistent Homology

PH is a computational topology method that quantifies the changes in the topological invariants of a Vietoris-Rips complex as a scale parameter $\rho$ is varied. In this section, we introduce some key concepts of PH. Further details on PH can be found in Refs. [20, 56].

**Notation.** $\mathcal{X} := \{x_i\}_{i=1}^n$ represents a point cloud and $\mu : \mathcal{X} \times \mathcal{X} \rightarrow \mathbb{R}$ denotes a distance metric over $\mathcal{X}$. Matrix $\mathcal{M}$ represents the pairwise distances (*i.e.*, Euclidean distance) between points in $\mathcal{X}$.

**Vietoris-Rips Complex.** The Vietoris-Rips complex [57] is a special simplicial complex constructed from a set of points in a metric space, and it can be used to approximate the topology of the underlying space. For $0 \leq \rho < \infty$, we represent the Vietoris-Rips complex of point cloud $\mathcal{X}$ at scale $\rho$ as $\mathcal{V}_\rho(\mathcal{X})$, which contains all simplices (*i.e.*, subsets) of $\mathcal{X}$, and each component of point cloud $\mathcal{X}$ satisfies a distance constraint: $\mu(x_i, x_j) \leq \rho$ for any $i, j$. Moreover, the Vietoris-Rips complex satisfies a nesting relation, *i.e.*, $\mathcal{V}_{\rho_i} \subseteq \mathcal{V}_{\rho_j}$ for any $\rho_i \leq \rho_j$, which allows us to track the evolution progress of simplical complex as the scale $\rho$ increases. It is worth noting that $\mathcal{V}_\rho(\mathcal{X})$ and $\mathcal{V}_\rho(\mathcal{M})$ are equivalent because constructing the Vietoris-Rips complex only requires distance.

**Homology Group.** The homology group [58] is an algebraic structures that analyzes the topological features of a simplicial complex in different dimension $j$, such as connected components ($H_0$), cycles ($H_1$), voids ($H_2$), and higher-dimensional features ($H_j, j \geq 3$). By tracking the changes in topological features ($H_j$) of the Vietoris-Rips complex as the scale $\rho$ increases, it is possible to gain insight into the multi-scale topological information of the underlying space.

**Persistence Diagram and Persistence Pairing [59].** The persistence diagram $\mathcal{D}$ is a multi-set of points $(b, d)$ in the Cartesian plane $\mathbb{R}^2$, which encodes information about the lifespan of topological features. Specially, it summarizes the birth time $b$ and death time $d$ of each topological feature, where birth time $b$ signifies the scale at which the feature is created and death time $d$ refers to the scale at which it is destroyed. The persistence pairing $\gamma$ contains indices $(i, j)$ corresponding to simplices $r_i, r_j \in \mathcal{V}_\rho(\mathcal{X})$ that create and destroy the topological features identified by $(b, d) \in \mathcal{D}$, respectively.

## 4 Methodology

In this paper, we propose a novel framework named **TopoFR** for constraining the FR model to preserve the topological structure information of the input space in their latent features. The architecture of our TopoFR model is depicted in Figure 3. It consists of two components: a feature extractor $\mathcal{F}$ and an image classifier $\mathcal{C}$. Mathematically, given an input image $x$, the latent feature extracted by $\mathcal{F}$ is denoted as $f = \mathcal{F}(x) \in \mathbb{R}^l$, and the classification probability predicted by $\mathcal{C}$ is denoted as $g = \mathcal{C}(f) \in \mathbb{R}^K$, where $l$ represents the feature dimension and $K$ denotes the number of classes. The entropy of the classification prediction probability $g$ can be represented as $E(g) = -\sum_{k=1}^K g^k \log g^k$, where $g^k$ is the probability of predicting a sample to class $k$.

## 4.1 Perturbation-guided Topological Structure Alignment

As mentioned in Section 1, directly applying PH to align the topological structures of the input space and the latent space can cause the FR model to encounter structure collapse phenomenon. To remedy this problem, we propose a Perturbation-guided Topological Structure Alignment (**PTSA**) strategy that includes two mechanisms: random structure perturbation and invariant structure alignment.

**Random Structure Perturbation (RSP).** PTSA first utilizes the RSP mechanism to randomly perturb the structure of the latent space. Specially, it utilizes a data augmentation list $\mathcal{A} = \{\mathcal{A}_1, \mathcal{A}_2, \mathcal{A}_3, \mathcal{A}_4\}$ that includes four common data augmentation operations, namely Random Erasing $\mathcal{A}_1$, GaussianBlur $\mathcal{A}_2$, Grayscale $\mathcal{A}_3$ and ColorJitter $\mathcal{A}_4$. For each training sample $x_i$, RSP will randomly select an operation $\mathcal{A}_r$ from $\mathcal{A}$ to perturb it, *i.e.*, $\widetilde{x}_i = \mathcal{A}_r(x_i)$. Then the perturbed sample $\widetilde{x}_i$ will be fed into the model for supervised learning, which effectively increases the structure diversity of the latent space. In our model, we adopt ArcFace loss [1] as the basic classification loss:

$$\mathcal{L}_{arc}(\widetilde{x}_i, y_i) = -\log \frac{e^{s(\cos(\theta_i^y + m))}}{e^{s(\cos(\theta_i^y + m))} + \sum_{k=1, k \neq y}^{K} e^{s \cos \theta_i^k}}, \tag{1}$$

where $y_i$ is the class label of the original image $x_i$, $s$ is a scaling parameter, $\theta_i^k$ is the angle between the $k$-th class center and feature, and $m$ denotes an additive angular margin. During training, we apply the RSP mechanism to each sample $x_i$ with a probability $\xi$ of 0.2.

**Invariant Structure Alignment (ISA).** Given a mini-batch of original training samples $\mathcal{X} = \{x_i\}_{i=1}^{n}$, we denote the perturbed batch samples as $\widetilde{\mathcal{X}} = \{\widetilde{x}_i\}_{i=1}^{n}$. For the perturbed samples, we denote the latent features extracted by $\mathcal{F}$ as $\widetilde{\mathcal{Z}} = \left\{\widetilde{f}_i\right\}_{i=1}^{n}$. During forward propagation, we can construct the Vietoris-Rips complexes $\mathcal{V}_\rho(\mathcal{X})$ and $\mathcal{V}_\rho(\widetilde{\mathcal{Z}})$ for point clouds $\mathcal{X}$ and $\widetilde{\mathcal{Z}}$ respectively, based on their respective pairwise distance matrix $\mathcal{M}^{\mathcal{X}}$ and $\mathcal{M}^{\widetilde{\mathcal{Z}}}$. Then we can utilize persistent homology to analyze the topological structures of $\mathcal{V}_\rho(\mathcal{X})$ and $\mathcal{V}_\rho(\widetilde{\mathcal{Z}})$, and obtain their corresponding persistence diagrams $\left\{\mathcal{D}^{\mathcal{X}}, \mathcal{D}^{\widetilde{\mathcal{Z}}}\right\}$ and persistence pairings $\left\{\gamma^{\mathcal{X}}, \gamma^{\widetilde{\mathcal{Z}}}\right\}$, respectively.

Ideally, no matter how the face image is perturbed, the position of the encoded face feature in the latent space should remain unchanged, and the topological structure of the perturbed latent space should also be consistent with the original input space. To this end, we choose to align the original input space $\mathcal{X}$ with the perturbed latent space $\widetilde{\mathcal{Z}}$ to achieve this goal. Prior studies usually utilize bottleneck distance or Wasserstein distance to measure the topological structure discrepancy [60, 18] between two spaces by comparing the differences in persistence diagrams. However, these two metrics are sensitive to outliers in persistence diagrams [55, 61] and will significantly increase models' training time, rendering them unsuitable for FR tasks with extremely large-scale datasets, as verified in Table 8 in the Appendix.

To mitigate this issue, we turn to retrieve the persistence diagrams values by subsetting the corresponding pairwise distance matrix with edge indices provided by the persistence pairings [62, 63, 64], *i.e.*, $\mathcal{D}^{\mathcal{X}} \simeq \mathcal{M}^{\mathcal{X}}[\gamma^{\mathcal{X}}]$ and $\mathcal{D}^{\mathcal{Z}} \simeq \mathcal{M}^{\widetilde{\mathcal{Z}}}[\gamma^{\widetilde{\mathcal{Z}}}]$. By comparing the difference between two topologically relevant distance matrices from both spaces, we can quickly and stably compute the discrepancy between their persistence diagrams, providing an efficient solution for structure alignment of FR models driven by large-scale datasets. We formulate the ISA loss as follows:

$$\mathcal{L}_{sa}(\mathcal{D}^{\mathcal{X}}, \mathcal{D}^{\widetilde{\mathcal{Z}}}) = \frac{1}{2} \left( \left\| \mathcal{M}^{\mathcal{X}}[\gamma^{\mathcal{X}}] - \mathcal{M}^{\widetilde{\mathcal{Z}}}[\gamma^{\mathcal{X}}] \right\|^2 + \left\| \mathcal{M}^{\widetilde{\mathcal{Z}}}[\gamma^{\widetilde{\mathcal{Z}}}] - \mathcal{M}^{\mathcal{X}}[\gamma^{\widetilde{\mathcal{Z}}}] \right\|^2 \right) \tag{2}$$

**Notably**, in the field of FR, most existing works do not include any data augmentation operations, as this would introduce more unidentifiable face images (*i.e.,* destroying the fidelity of each face), which generally hurts the FR model's generalization ability, as verified in Refs. [3, 65]. In this work, we do not employ these data augmentations to simply augment data scale. Instead, we use them to increase the latent space's structure diversity, effectively addressing the structure collapse problem. As a result, our PTSA strategy can reap the benefits of data augmentations while mitigating their potential negative effects (see Figure 2, Table 3, and Figure 5 for further analysis.).

## 4.2 Structure Damage Estimation

In practical FR scenarios, low-quality face samples, also known as "hard samples", are commonly included in the training set. These hard samples tend to be encoded in abnormal positions near the decision boundary in the latent space [25, 6, 66, 67], which will disrupt the latent space's topological structure and further hinder the alignment of structures. To address this issue, we propose a novel hard sample mining strategy called Structure Damage Estimation (**SDE**). SDE is specifically designed to identify hard samples with serious structure damage within the training set accurately. By prioritizing the learning of these hard samples and guiding them back to the reasonable positions during optimization, SDE aims to mitigate the adverse impact of hard samples on the latent space's topological structure.

**Prediction Uncertainty.** Hard samples are typically distributed near the decision boundary, thus have a high prediction uncertainty (*i.e.*, large entropy of the classifier prediction) and are more likely to be misclassified by the classifier [68, 69, 70, 71]. Conversely, easy samples are usually located far from the decision boundary and have relatively low prediction uncertainty. To model the difficulty of each sample, we introduce a binary random variable $u_i \in \{0, 1\}$ for each sample $\widetilde{x}_i$ to indicate whether the sample is hard or easy by values of 1 and 0, respectively. Then the probability that sample $\widetilde{x}_i$ belongs to hard samples (*i.e.*, with large prediction uncertainty) can be defined as $h_\varphi(\widetilde{x}_i) = P_\varphi(u_i = 1|\widetilde{x}_i)$, where $\varphi$ represents the parameter set. According to the cluster assumption [72, 73], we believe that samples with higher prediction entropy are more disruptive to the latent space's topological structure. Therefore, we propose to model the distribution of the entropy $E(\widetilde{g}_i)$ for each training sample $\widetilde{x}_i$ using the Gaussian-uniform mixture (**GUM**) model, a statistical distribution that is robust to outliers [74, 75, 76]:

$$p\left(E(\widetilde{g}_i)|\widetilde{x}_i\right) = \pi \mathcal{N}^+(E(\widetilde{g}_i)|0, \Sigma) + (1 - \pi)\mathcal{U}(0, \Omega), \tag{3}$$

where

$$\mathcal{N}^+(a|0, \Sigma) = \begin{cases} 2\mathcal{N}(a|0, \Sigma), & a \geq 0. \\ 0, & a < 0. \end{cases} \tag{4}$$

$\mathcal{U}(0, \Omega)$ is a uniform distribution defined on $[0, \Omega]$, $\pi$ is a prior probability, and $\Sigma$ is the variance of Gaussian distribution $\mathcal{N}(a|0, \Sigma)$. In this mixed model, the uniform distribution term $\mathcal{U}$ and the Gaussian distribution term $\mathcal{N}^+$ respectively model the hard samples and easy samples. Then the posterior probability that the sample $\widetilde{x}_i$ to be hard (*i.e.*, high-uncertainty) can be computed as follows:

$$h_\varphi(\widetilde{x}_i) = P_\varphi(u_i = 1|\widetilde{x}_i) = \frac{(1 - \pi)\mathcal{U}(0, \Omega)}{\pi \mathcal{N}^+(E(\widetilde{g}_i)|0, \Sigma) + (1 - \pi)\mathcal{U}(0, \Omega)}. \tag{5}$$

In Equation (5), when the classifier prediction is close to uniform distribution or when the prediction probabilities for multiple classes are nearly equal, the posterior probability of a sample belonging to hard data will be very high, *i.e.*, $(\widetilde{g}_i \to [\frac{1}{K}, \frac{1}{K}, \cdots, \frac{1}{K}], h_\varphi(\widetilde{x}_i) \to 1)$, otherwise it is relatively low. Hence, the prediction uncertainty of sample $x_i$ can be measured by a quantitative probability $h_\varphi(\widetilde{x}_i)$.

Assume $\widehat{E}(\widetilde{g}_i) = (-1)^{\epsilon_i} E(\widetilde{g}_i)$, $\epsilon_i \sim B(1, 0.5)$, where $B$ is a Bernoulli distribution [77], then the variable $\widehat{E}(\widetilde{g}_i)$ obeys the following statistical distribution:

$$p\left(\widehat{E}(\widetilde{g}_i)|\widetilde{x}_i\right) = \pi \mathcal{N}(\widehat{E}(\widetilde{g}_i)|0, \Sigma) + (1 - \pi)\mathcal{U}(-\Omega, \Omega). \tag{6}$$

In this way, the maximum likelihood model of Equation (6) can be formulated as: $\max\limits_{\pi, \Sigma, \Omega} \prod\limits_{i=1}^{n} p\left(\widehat{E}(\widetilde{g}_i)|\widetilde{x}_i\right)$. Then, the parameter set $\varphi = \{\pi, \Sigma, \Omega\}$ of GUM can be estimated via the Expectation-Maximization (EM) algorithm [78] with the following iterative formulas:

$$h_\varphi^{(t+1)}(\widetilde{x}_i) = \frac{(1 - \pi^{(t)})\mathcal{U}(-\Omega^{(t)}, \Omega^{(t)})}{\pi^{(t)}\mathcal{N}(\widehat{E}(\widetilde{g}_i)|0, \Sigma^{(t)}) + (1 - \pi^{(t)})\mathcal{U}(-\Omega^{(t)}, \Omega^{(t)})}, \pi^{(t+1)} = \frac{\sum_{i=1}^{n}(1 - h_\varphi^{(t+1)}(\widetilde{x}_i))}{n},$$

$$\Sigma^{(t+1)} = \frac{\sum_{i=1}^{n}(1 - h_\varphi^{(t+1)}(\widetilde{x}_i))(\widehat{E}(\widetilde{g}_i))^2}{\sum_{i=1}^{n}(1 - h_\varphi^{(t+1)}(\widetilde{x}_i))}, \Omega^{(t+1)} = \sqrt{3(\eta_2 - \eta_1^2)}, \tag{7}$$

where

$$\eta_1 = \frac{\sum_{i=1}^{n} \frac{h_{\varphi}^{(t+1)}(\widetilde{x}_i)}{1-\pi^{(t+1)}} \widehat{E}(\widetilde{g}_i)}{\sum_{i=1}^{n}(1 - h_{\varphi}^{(t+1)}(\widetilde{x}_i))}, \eta_2 = \frac{\sum_{i=1}^{n} \frac{h_{\varphi}^{(t+1)}(\widetilde{x}_i)}{1-\pi^{(t+1)}} (\widehat{E}(\widetilde{g}_i))^2}{\sum_{i=1}^{n}(1 - h_{\varphi}^{(t+1)}(\widetilde{x}_i))}.$$

Specifically, at each iteration, EM alternates between evaluating the expected log-likelihood (E-step) and updating the parameter set $\varphi$ (M-step). In Equation (7), the **E-step** aims to evaluate the posterior probability $h_{\varphi}^{(t+1)}$ of an sample $\widetilde{x}_i$ to be hard sample using the iterative formula $h_{\varphi}^{(t+1)}(\widetilde{x}_i)$, where $(t+1)$ denotes the EM iteration index. The **M-step** updates the parameter set $\varphi$ using the iterative formulas $\pi^{(t+1)}$, $\Sigma^{(t+1)}$ and $\Omega^{(t+1)}$.

**Structure Damage Score (SDS).** Compared to correctly classified samples, misclassified samples usually have larger difficulty and have greater destructive effects on the latent space's topological structure. Therefore, misclassified samples need to receive more attention during training. Inspired by the Focal loss [79], we design a probability-aware scoring mechanism $\omega(\widetilde{x}_i)$ that combines prediction uncertainty $h_{\varphi}(\cdot)$ and prediction accuracy to adaptively compute SDS for each sample $\widetilde{x}_i$:

$$\omega(\widetilde{x}_i) = \omega_1(\widetilde{x}_i) \times \omega_2(\widetilde{x}_i) = (1 + h_{\varphi}(\widetilde{x}_i))^{\lambda} \times (1 - \widetilde{g}_i^{gt}), \tag{8}$$

where $\lambda$ is a temperature coefficient, and $\widetilde{g}_i^{gt}$ represents the prediction probability of ground truth. Specifically, SDE assigns higher SDS to hard samples and lower SDS to easy samples, which effectively balances the contribution of each sample to the objective. By assigning higher scores to hard samples, the model is encouraged to focus more on learning these challenging samples, boosting the FR system's generalization. Formally, the SDS weighted classfication loss $\mathcal{L}_{cls}$ is defined as:

$$\mathcal{L}_{cls} = \omega(\widetilde{x}_i) \times \mathcal{L}_{arc}(\widetilde{x}_i, y_i) \tag{9}$$

During training, to minimize the objective $\mathcal{L}_{cls}$, the model needs to optimize both the SDS $\omega$ and the loss $\mathcal{L}_{arc}$, which brings two benefits: **(1)** Minimizing $\mathcal{L}_{arc}$ can encourage the model to capture face features with greater generalization ability from diverse training samples. **(2)** Minimizing SDS $\omega$ can alleviate the damage of hard samples to the latent space's topological structure, which is beneficial to the preservation of topological structure information and the construction of clear decision boundary.

### 4.3 Model Optimization

To summarize, the overall objective of TopoFR can be formulated as follows:

$$\min_{\mathcal{F},\mathcal{C}} \mathcal{L}_{cls} + \alpha \mathcal{L}_{sa} \tag{10}$$

where $\alpha$ is hyper-parameter that balances the contributions of the loss $\mathcal{L}_{cls}$ and the loss $\mathcal{L}_{sa}$. Detailed parameter sensitivity analysis can be found in Figure 6 in the Appendix.

## 5 Experiments

### 5.1 Datasets.

**i) For training**, we employ three distinct datasets, namely MS1MV2 [1] (5.8M facial images, 85K identities), Glint360K [41] (17.1M facial images, 360K identities), and WebFace42M [80] dataset (42.5M facial images, 2M identities). **ii) For evaluation**, we adopt LFW [81], AgeDB-30 [82], CFP-FP [83], CPLFW [84], CALFW [85], IJB-C [23], IJB-B [86] and the ICCV-2021 Masked Face Recognition Challenge (**MFR-Ongoing**) [27] as the benchmarks to test the performance of our models.

Notably, the MFR-Ongoing [27] is the most authoritative and comprehensive competition for evaluating FR models' generalization performance. It contains not only the existing popular test sets, such as IJB-C, but also its own MFR benchmarks, such as Mask, Children, and Multi-Racial test sets. Due to page size limitation, more training settings and experimental results are placed on **Appendix**.

### 5.2 Results on Mainstream Benchmarks

**Results on MFR-Ongoing.** We employ WebFace42M as training set, and compare our TopoFR with SOTA competitors on MFR-Ongoing challenge, as reported in Table 1. For a fair comparison, all

Table 1: Verification accuracy (%) on the MFR-Ongoing benchmark.

| Method | Training Data | Venue | MFR | | | | | | | IJB-C | |
| | | | Mask | Children | African | Caucasian | South Asian | East Asian | MR-All | 1e-5 | 1e-4 |
|---|---|---|---|---|---|---|---|---|---|---|---|
| R200, Partial FC [43] | | CVPR22 | 91.87 | - | 97.79 | 98.70 | 98.54 | 89.52 | 97.70 | 96.93 | 97.97 |
| R200, UniFace [87] | WebFace42M | ICCV23 | 92.43 | 93.11 | **98.14** | **98.98** | 98.84 | 90.01 | 97.92 | 96.68 | 97.91 |
| **R200, TopoFR** | | NeurIPS24 | **93.96** | **93.57** | 97.97 | 98.71 | **98.98** | **92.85** | **98.13** | **97.10** | **98.01** |

Table 2: Verification accuracy (%) on LFW, CFP-FP, AgeDB-30, IJB-C and IJB-B benchmarks.

| Training Data | Method | Venue | LFW | CFP-FP | AgeDB-30 | IJB-C | | IJB-B |
| | | | | | | 1e-5 | 1e-4 | 1e-4 |
|---|---|---|---|---|---|---|---|---|
| MS1MV2 | R50, ArcFace [1] | CVPR19 | 99.68 | 97.11 | 97.53 | 88.36 | 92.52 | 91.66 |
| | R50, MagFace [5] | CVPR21 | 99.74 | 97.47 | 97.70 | 88.95 | 93.34 | 91.47 |
| | R50, AdaFace [3] | CVPR22 | 99.82 | 97.86 | 97.85 | - | 96.27 | 94.42 |
| | R50, **TopoFR**† | NeurIPS24 | **99.83** | **98.24** | 98.23 | **94.79** | 96.42 | 95.13 |
| | R50, **TopoFR** | NeurIPS24 | **99.83** | **98.24** | **98.25** | 94.71 | **96.49** | **95.14** |
| | R100, CosFace [2] | CVPR18 | 99.78 | 98.26 | 98.17 | 92.68 | 95.56 | 94.01 |
| | R100, ArcFace [1] | CVPR19 | 99.77 | 98.27 | 98.15 | 92.69 | 95.74 | 94.09 |
| | R100, MV-Softmax [88] | AAAI20 | 99.80 | 98.28 | 97.95 | - | 95.20 | 93.60 |
| | R100, URL [65] | CVPR20 | 99.78 | 98.64 | - | 95.00 | 96.60 | - |
| | R100, BroadFace [89] | ECCV20 | **99.85** | 98.63 | 98.38 | 94.59 | 96.38 | 94.97 |
| | R100, CurricularFace [4] | CVPR20 | 99.80 | 98.37 | 98.32 | - | 96.10 | 94.80 |
| | R100, MagFace+ [5] | CVPR21 | 99.83 | 98.46 | 98.17 | 94.08 | 95.97 | 94.51 |
| | R100, SCF-ArcFace [25] | CVPR21 | 99.82 | 98.40 | 98.30 | 94.04 | 96.09 | 94.74 |
| | R100, DAM-CurricularFace [90] | ICCV21 | - | - | - | - | 96.20 | 95.12 |
| | R100, ElasticFace-Cos+ [91] | CVPR22 | 99.80 | 98.73 | 98.28 | - | 96.65 | 95.43 |
| | R100, AdaFace [3] | CVPR22 | 99.82 | 98.49 | 98.05 | - | 96.89 | 95.67 |
| | TransFace-B [9] | ICCV23 | 99.82 | 98.39 | 98.27 | 94.15 | 96.55 | - |
| | R100, **TopoFR**† | NeurIPS24 | **99.85** | **98.83** | **98.42** | **95.28** | **96.96** | **95.70** |
| | R100, **TopoFR** | NeurIPS24 | **99.85** | 98.71 | **98.42** | 95.23 | 96.95 | **95.70** |
| | R200, ArcFace [1] | CVPR19 | 99.79 | 98.44 | 98.19 | 94.67 | 96.53 | 95.18 |
| | R200, AdaFace [3] | CVPR22 | 99.83 | 98.76 | 98.28 | 94.88 | 96.93 | 95.71 |
| | TransFace-L [9] | ICCV23 | 99.83 | 98.65 | 98.23 | 94.55 | 96.59 | - |
| | R200, **TopoFR**† | NeurIPS24 | **99.85** | **99.09** | **98.54** | **95.19** | **97.12** | 95.77 |
| | R200, **TopoFR** | NeurIPS24 | **99.85** | 99.05 | 98.52 | 95.15 | 97.08 | **95.82** |
| Glint360K | R50, ArcFace [1] | CVPR19 | 99.78 | 98.77 | 98.28 | 95.29 | 96.81 | 95.30 |
| | R50, AdaFace [3] | CVPR22 | 99.82 | 99.07 | 98.34 | 95.58 | 96.90 | 95.66 |
| | R50, **TopoFR** | NeurIPS24 | **99.85** | **99.28** | **98.47** | **95.99** | **97.27** | **95.96** |
| | R100, ArcFace [1] | CVPR19 | 99.81 | 99.04 | 98.31 | 95.38 | 96.89 | 95.69 |
| | R100, AdaFace [3] | CVPR22 | 99.82 | 99.20 | 98.58 | 96.24 | 97.19 | 95.87 |
| | TransFace-B [9] | ICCV23 | 99.85 | 99.17 | 98.53 | 96.18 | 97.45 | - |
| | R100, **TopoFR** | NeurIPS24 | **99.85** | **99.43** | **98.72** | **96.57** | **97.60** | **96.34** |
| | R200, ArcFace [1] | CVPR19 | 99.82 | 99.14 | 98.49 | 95.71 | 97.20 | 95.89 |
| | R200, AdaFace [3] | CVPR22 | 99.83 | 99.24 | 98.61 | 95.96 | 97.33 | 96.12 |
| | TransFace-L [9] | ICCV23 | 99.85 | 99.32 | 98.62 | 96.29 | 97.61 | - |
| | R200, **TopoFR** | NeurIPS24 | **99.87** | **99.45** | **98.82** | **96.71** | **97.84** | **96.56** |

compared models adopt ResNet-200 as the backbone. Specially, our TopoFR surpasses the SOTA competitors UniFace and Partial FC in multiple metrics, implying the superiority of our method. Until the submission of this work (May 22 '24), the proposed TopoFR **ranks second place** on the academic track of the MFR-Ongoing leaderboard: `http://iccv21-mfr.com/#/leaderboard/academic`.

**Results on LFW, CFP-FP and AgeDB-30.** We adopt MS1MV2 and Glint360K to train our models, respectively. The results are reported in Table 2. To showcase the universality of our method, we also provide detailed experimental results of **TopoFR**† model trained by CosFace [2]. As stated in Refs.[3, 5], the performances of existing FR models on these three benchmarks have reached saturation. **1)** On MS1MV2 training set, we note that our TopoFR and TopoFR† models still obtain accuracy improvement and outperform SOTA competitors (*e.g.*, AdaFace [3] and TransFace [9]). **2)** On Glint360K training set, our TopoFR become SOTA models and surpass AdaFace and TransFace.

**Results on IJB-C and IJB-B.** We train our TopoFR on MS1MV2 and Glint360K respectively, and compare with SOTA methods on IJB-C and IJB-B benchmarks, as reported in Table 2. **1)** On MS1MV2 training set, our models obtain the best results under different backbones. For instance, our R50 TopoFR and R50 TopoFR† models greatly surpass SOTA method AdaFace and even beat most R100-based competitors. **2)** On Glint360K training set, all our models significantly outperform the cutting-edge competitor AdaFace and achieve SOTA performance. More importantly, our TopoFR even works better than ViT-based SOTA method TransFace, implying the superiority of our method.

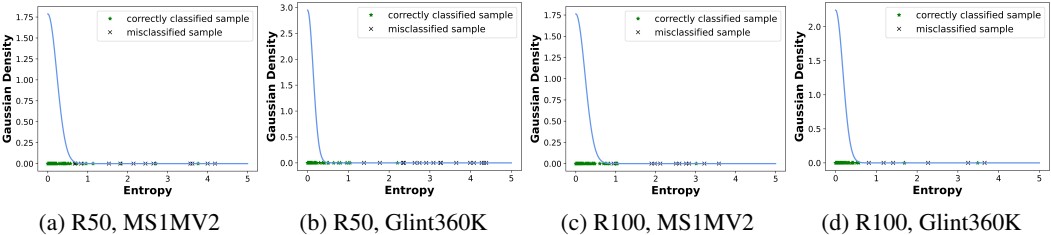

(a) R50, MS1MV2     (b) R50, Glint360K     (c) R100, MS1MV2     (d) R100, Glint360K

Figure 4: The estimated Gaussian density (blue curve) *w.r.t* the entropy of classification prediction. Green marker $\star$ and black marker $\times$ represent the entropy of correctly classified sample and misclassified sample, respectively.

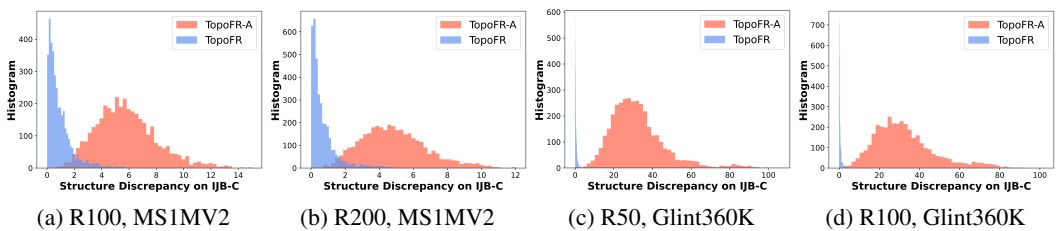

(a) R100, MS1MV2     (b) R200, MS1MV2     (c) R50, Glint360K     (d) R100, Glint360K

Figure 5: The topological structure discrepancy of TopoFR and variant TopoFR-A under different backbones and training datasets (*i.e.*, **[Backbone, Training dataset]**). Variant TopoFR-A directly utilizes PH to align the topological structures of two spaces. Notably, our TopoFR models trained with Glint360K dataset almost perfectly align the topological structures of the input space and the latent space on the IJB-C benchmark (*i.e.*, the blue histogram almost converges to a straight line).

## 5.3 Analysis and Ablation Study

Due to the limitation of page size, more ablation experiments and analysis are placed on **Appendix**.

**1) Contribution of Each Component:** To investigate the contribution of each component in our model, we employ MS1MV2 as the training set, and compare ArcFace (baseline), and four variants of TopoFR on the IJB-C benchmark. The variants of TopoFR are as follows: (1) **TopoFR-R**, the variant only adds RSP mechanism to ArcFace. (2) **TopoFR-A**, based on ArcFace, the variant simply aligns the structure of input space and latent space without using RSP. (3) **TopoFR-P**, the variant fully introduces the PTSA strategy into ArcFace. (4) **TopoFR-G**, based on TopoFR-P, the variant only uses prediction uncertainty $\omega_1$ modeled by GUM to re-weight each sample. (5) **TopoFR-F**, based on TopoFR-P, the variant simply applies Focal loss $\omega_2$ to re-weight each sample.

Table 3: Ablation study.

| Training Data | Method | IJB-C(1e-4) |
|---|---|---|
| MS1MV2 | R50, ArcFace | 92.52 |
| | R50, TopoFR-R | 92.44 |
| | R50, TopoFR-A | 93.26 |
| | R50, TopoFR-P | 95.34 |
| | R50, TopoFR-F | 95.40 |
| | R50, TopoFR-G | 96.23 |
| | R50, TopoFR | **96.49** |

The results gathered in Table 3 reflect some observations: (1) Compared with ArcFace, the accuracy of TopoFR-R is clearly reduced due to the addition of more unidentifiable face images, which hurts the FR model's generalization ability. (2) TopoFR-A outperforms ArcFace, indicating that directly aligning the two spaces can slightly boost model's performance, but it inevitably encounters structure collapse issue. (3) TopoFR-P greatly surpasses TopoFR-R and TopoFR-A, implying that preserving the structure information can greatly improve FR model's generalization. (4) TopoFR outperforms TopoFR-F and TopoFR-G, which not only demonstrates the effectiveness of SDE strategy, but also indicates that the prediction uncertainty $\omega_1$ is complementary to Focal loss $\omega_2$ in mining hard samples.

**2) Effectiveness of GUM:** To visually demonstrate the effectiveness of GUM in mining hard samples, we present the estimated Gaussian density of the prediction entropy during training in Figure 4. These curves show that the entropy of misclassified face samples (represented by black crosses) usually have rather low Gaussian density (*i.e.*, high posterior probability $h_\varphi$), thus can be easily detected.

Note that even if some misclassified samples have small entropy (*i.e.*, high Gaussian density and low posterior probability $h_\varphi$), their SDS $\omega$ can still be corrected by the Focal loss $\omega_2$.

**3) Generalization of PTSA:** To show the superior generalization ability of our PTSA strategy in preserving structure information, we investigate the topological structure discrepancy between the input and the latent spaces of TopoFR and its variant TopoFR-A on IJB-C benchmark. Note that TopoFR-A directly utilizes PH to align the topological structures of two spaces. The results in Figure 5 indicate that: 1) Directly using PH to align the topological structures of two spaces does not effectively reduce the structure discrepancy, as the model encounters the structure collapse issue; 2) PTSA strategy can effectively align the topological structures of two spaces and address this structure collapse issue. Remarkably, Figures 5c and 5d show that our TopoFR models trained on Glint360K almost perfectly preserve structure information of input spaces in their latent features, thereby verifying the generalization of PTSA strategy.

# 6   Conclusion

This paper proposes a novel FR framework called TopoFR that aims to encode the critical structure information in large-scale face dataset into the latent space. Specially, TopoFR leverages a structure alignment strategy PTSA and a hard sample mining strategy SDE. PTSA employs PH to reduce the topological structure discrepancy between the input and latent spaces, effectively mitigating structure collapse phenomenon and preserving structure information. SDE accurately identifies hard samples and guides the model to prioritize optimizing these samples, mitigating their adverse impact on the latent space's structure. Comprehensive experiments validate the superiority of our TopoFR.

# 7   Broader Impacts

It would be good to mention that the utilization of face images do not have any privacy concern given the datasets have proper license and users consent to distribute biometric data for research purpose. We address the well-defined face recognition task and conduct experiments on publicly available face datasets. Therefore, the propose method does not involve sensitive attributes and we do not notice any negative societal issues.

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

# A  Appendix

## A.1  Implementation Details

**Training Details.** For MS1MV2 and Glint360K, our models are trained using Pytorch on 4 NVIDIA Tesla A100 GPUs, and a mini-batch of 128 images is assigned for each GPU. In the case of WebFace42M, we train our models (ResNet-200 backbone) using 64 NVIDIA Tesla V100 GPUs. We crop all images to 112×112, following the same setting as in ArcFace [1, 34]. For the backbones, we adopt ResNet-50, ResNet-100 and ResNet-200 [92] as modified in [1]. We follow [1] to employ ArcFace ($s = 64$ and $m = 0.5$) as the basic classification loss to train the TopoFR model. For the TopoFR$^\dagger$ model trained by CosFace [2], we set the scale $s$ to 64 and the cosine margin $m$ of CosFace to 0.4. To optimize the models, we use Stochastic Gradient Descent (SGD) optimizer with momentum of 0.9 for both datasets. The weight decay for MS1MV2 is set to 5e-4 and 1e-4 for Glint360K. The initial learning rate is set to 0.1 for both datesets. In terms of the balance coefficient $\alpha$, we choose $\alpha = 0.1$ for experiments on R50 TopoFR, and $\alpha = 0.05$ for experiments on R100 TopoFR and R200 TopoFR. During training, we apply RSP mechanism with a certain probability. Specially, for an original input sample $x$, the probability of it undergoing RSP is $\xi$, and the probability of it remaining unchanged is $1 - \xi$. For the hyper-parameter $\xi$, we choose $\xi = 0.2$. Notably, in our method, we focus on preserving the 0-dimension homology $H_0$ in the topological structure alignment loss $\mathcal{L}_{sa}$. Because preliminary experiments demonstrated that using the 1-dimension or higher-dimension homology only increases model's training time without clear performance gains. Code and pre-trained models are available at: `https://github.com/modelscope/facechain/tree/main/face_module/TopoFR`.

**Evaluation Protocol on MFR-Ongoing Challenge.** In the MFR-Ongoing Challenge, the trained models are submitted to and evaluated by the online server. Specially, "TAR@FAR=1e-4" and "TAR@FAR=1e-5" are reported on the IJB-C. Furthermore, we report "TAR@FAR=1e-4" for Mask and Children test sets, and "TAR@FAR=1e-6" for MR-ALL test set. The academic track of the ongoing MFR challenge leaderboard can be found on `http://iccv21-mfr.com/#/leaderboard/academic`. More details about the MFR-Onoging Challenge can be found in Ref. [27].

## A.2  More Results

### A.2.1  Experiments on Other Benchmarks

**Results on CPLFW and CALFW.** We utilize MS1MV2 as the training set, and compare our TopoFR with SOTA methods on CPLFW and CALFW benchmarks, as reported in 4. As can be seen, the proposed R50 TopoFR and R50 TopoFR$^\dagger$ greatly outperform the SOTA competitor R50 AdaFace on two benchmarks. Furthermore, under the ResNet-100 backbone, our TopoFR also achieves SOTA performance and surpasses AdaFace by 0.26% and 0.36% on CPLFW and CALFW respectively.

Table 4: Verification accuracy (%) on CPLFW and CALFW.

| Training Data | Method | Venue | CPLFW | CALFW |
|---|---|---|---|---|
| | R50, ArcFace [1] | CVPR19 | 91.76 | 95.14 |
| | R50, MagFace [5] | CVPR21 | 92.49 | 95.88 |
| | R50, AdaFace [3] | CVPR22 | 92.83 | 96.07 |
| | R50, **TopoFR**$^\dagger$ | NeurIPS24 | 93.36 | 96.24 |
| | R50, **TopoFR** | NeurIPS24 | **93.38** | **96.25** |
| | R100, CosFace [2] | CVPR18 | 92.26 | 95.75 |
| | R100, ArcFace [1] | CVPR19 | 92.10 | 95.47 |
| | R100, MV-Softmax [88] | AAAI20 | 92.83 | 96.10 |
| MS1MV2 | R100, BroadFace [89] | ECCV20 | 93.17 | 96.20 |
| | R100, CurricularFace [4] | CVPR20 | 93.13 | 96.20 |
| | R100, MagFace [5] | CVPR21 | 92.87 | 96.15 |
| | R100, SCF-ArcFace [25] | CVPR21 | 93.16 | 96.12 |
| | R100, ElasticFace-Arc+ [91] | CVPR22 | 93.28 | 96.17 |
| | R100, ElasticFace-Cos+ [91] | CVPR22 | 93.23 | 96.18 |
| | R100, AdaFace [3] | CVPR22 | 93.53 | 96.08 |
| | R100, IIC-AdaFace [93] | ICLR24 | 93.48 | 96.18 |
| | R100, **TopoFR**$^\dagger$ | NeurIPS24 | 93.78 | 96.42 |
| | R100, **TopoFR** | NeurIPS24 | **93.79** | **96.44** |

### A.2.2 Experiments on Light-weight Network

To demonstrate the universality of our method, we conduct some experiments on a lightweight network MobileFaceNet-0.45G [94], as shown in Table 5. We can observe that with the help of SDE and PTSA strategies, the MobileFaceNet-0.45G model can achieve higher recognition accuracy, implying the effectiveness and universality of our method.

Table 5: Verification performance (%) on IJB-C. MobileFaceNet refers to the MobileFaceNet-0.45G backbone [94].

| Training Data | Method | IJB-C (1e-6) | IJB-C (1e-5) | IJB-C (1e-4) |
|---|---|---|---|---|
| MS1MV2 | MobileFaceNet | 81.75 | 90.13 | 93.42 |
| MS1MV2 | MobileFaceNet + PTSA + SDE | **83.14** | **91.01** | **94.48** |
| Glint360K | MobileFaceNet | 83.67 | 92.49 | 94.86 |
| Glint360K | MobileFaceNet + PTSA + SDE | **85.44** | **93.35** | **95.79** |

### A.3 More Ablation Experiments and Analysis

**1) Does Structure Information Improve Intra-class and Inter-class Relationships?:** We conduct two pairs of ablative experiments to validate the relationships between topological structure alignment and intra-class distance constraint as well as inter-class distance constraint. The results are gathered in Table 6.

We can find that integrating topological structure alignment with single inter-class or intra-class distance constraint can both obtain additional significant performance gains. This indicates that topological structure alignment can provide the extra structure information of intra-class and inter-class relationships, thereby establishing clearer decision boundaries. Overall, these improved results demonstrate topological structure alignment implicitly encourages intra-class compactness and inter-class separability in the deep feature space.

Table 6: Verification performance (%) on IJB-C. Relationships between Topological Structure Information and Intra-class/ Inter-class constraints.

| Training Data | Method | IJB-C (1e-6) | IJB-C (1e-5) | IJB-C (1e-4) |
|---|---|---|---|---|
| MS1MV2 | R100, ArcFace (w/o intra-class) | 82.37 | 89.13 | 94.16 |
| | R100, ArcFace(w/o intra-class) + topological constraint | **83.69** | **91.48** | **94.86** |
| MS1MV2 | R100, ArcFace(w/o inter-class) | 82.72 | 89.51 | 94.47 |
| | R100, ArcFace(w/o inter-class) + topological constraint | **84.53** | **92.36** | **95.12** |

**2) Comparison with Previous Hard Sample Mining Strategies:** To further demonstrate the superiority of our SDE strategy in mining hard samples, we compare it with existing hard sample mining strategies, including MV-Softmax [88], $AT_k$ [95] loss, Focal loss [79] and the recently proposed EHSM [9].

The results are gathered in Table 7. We can observe that the proposed SDE strategy clearly outperforms than previous hard sample mining strategies, indicating that our SDE strategy is better able to measure sample difficulty and improve the model's generalization performance. This is because the SDE comprehensively considers the prediction uncertainty and label information (*i.e.*, prediction probability of ground truth) when mining hard samples.

**3) Effectiveness of ISA:** Previous works often use Bottleneck distance and p-Wasserstein distance to measure the distance between persistence diagrams $\mathcal{D}^{\mathcal{X}}$ and $\mathcal{D}^{\widetilde{\mathcal{Z}}}$ [18, 60]. Concretely, the Bottleneck distance is defined as $\mathcal{L}_{\infty}(\mathcal{D}^{\mathcal{X}}, \mathcal{D}^{\widetilde{\mathcal{Z}}}) = \inf_{\kappa:\mathcal{D}^{\mathcal{X}} \to \mathcal{D}^{\widetilde{\mathcal{Z}}}} \sup_{\varpi \in \mathcal{D}^{\mathcal{X}}} \|\varpi - \kappa(\varpi)\|_{\infty}$, with $\kappa$ ranging over all bijections between sets of persistent intervals in diagrams $\mathcal{D}^{\mathcal{X}}$ and $\mathcal{D}^{\widetilde{\mathcal{Z}}}$, and $\|\cdot\|_{\infty}$ denotes the $\infty-$norm. Equivalently, the p-Wasserstein distance is defined as $\mathcal{L}_p(\mathcal{D}^{\mathcal{X}}, \mathcal{D}^{\widetilde{\mathcal{Z}}}) = (\inf_{\kappa:\mathcal{D}^{\mathcal{X}} \to \mathcal{D}^{\widetilde{\mathcal{Z}}}} \sum_{\varpi \in \mathcal{D}^{\mathcal{X}}} \|\varpi - \kappa(\varpi)\|_{\infty}^{p})^{1/p}$. However, the Bottleneck distance metric and p-Wasserstein distance metric are sensitive to outliers [55, 61, 96] and will significantly increase the training time of FR models. This makes them unsuitable for FR tasks with extremely large-scale datasets.

Table 7: Comparison with Previous Hard Sample Mining Strategies.

| Training Data | Method | IJB-C (1e-6) | IJB-C (1e-5) | IJB-C (1e-4) |
|---|---|---|---|---|
| MS1MV2 | R100, ArcFace | 85.65 | 92.69 | 95.74 |
| | R100, ArcFace + AT$_k$ | 85.93 | 93.04 | 95.89 |
| | R100, ArcFace + MV-Softmax | 86.47 | 93.38 | 95.94 |
| | R100, ArcFace + Focal Loss | 87.58 | 93.87 | 96.11 |
| | R100, ArcFace + EHSM | 88.30 | 94.19 | 96.18 |
| | R100, ArcFace + **SDE** | **89.23** | **94.65** | **96.49** |

Table 8: Comparison of TopoFR using Different Metrics.

| Training Data | Method | Average Training Time / Epoch | IJB-C (1e-4) |
|---|---|---|---|
| MS1MV2 | R100, TopoFR-B | 4312.56s | 96.76 |
| | R100, TopoFR-W | 4127.29s | 96.81 |
| | R100, **TopoFR** | **2729.28s** | **96.95** |

To demonstrate the stability and computational efficiency of our ISA strategy, we employ MS1MV2 as the training set, and compare TopoFR and its two variants on the IJB-C benchmark. The variants of TopoFR are as follows: (1) **TopoFR-B**, the variant uses Bottleneck distance to compute the discrepancy between persistence diagrams. (2) **TopoFR-W**, the variant adopts 1-Wasserstein distance to measure the discrepancy between persistence diagrams.

We provide the average training time per epoch of these models and their recognition performance on "TAR@FAR=1e-4" of the IJB-C benchmarks. The results presented in Table 8 reflect the following observations: **(1)** TopoFR outperforms TopoFR-B and TopoFR-W on "TAR@FAR=1e-4", indicating the robustness of our ISA to outliers. **(2)** Compared with TopoFR-B and TopoFR-W, our proposed TopoFR has a shorter training time, demonstrating the computational efficiency of ISA.

**4) Effect of Batch Size:** We investigate the model's performance under different batch size, and the results are gathered in Table 9.

We find that increasing the batch size does not lead to a significant improvement in model's recognition accuracy. Additionally, larger batch size imposes a greater workload on GPUs. Therefore, to strike a balance between model's accuracy and and GPU computational load, we choose to set the batch size to 128 in our TopoFR model.

Table 9: The Performance of TopoFR Model Under Different Batch Size.

| Training Data | Method | Batch Size | IJB-C (1e-5) | IJB-C (1e-4) |
|---|---|---|---|---|
| MS1MV2 | R100, TopoFR | 128 | **95.23** | **96.95** |
| | R100, TopoFR | 256 | 95.22 | 96.91 |
| | R100, TopoFR | 512 | 95.20 | 96.93 |

**5) Training Time:** For detailed training time analysis, please refer to the Table 10. Due to the introduction of the structure alignment strategy PTSA and hard sample mining strategy SDE, our TopoFR models require longer training time (1.16x). Specially, compared to the vanilla R50 ArcFace model, our R50 TopoFR model requires about 2 seconds more training time per 100 steps, which does not significantly increase the training time but brings a large performance gain. And our R100 topoFR model requires about 3 seconds extra training time per 100 steps than the vanilla R100 ArcFace model, which is not a major increase in traing time but leads to a significant performance advantage.

While for inference computation head, our method performs consistently with that of vanilla ArcFace model, since we adopt the same network architecture and data pre-process module.

Moreover, the results in Table 10 indicate that introducing SDE strategy (*i.e.*, GUM) leads to significant performance improvements with only a small increase in training time (*i.e.*, R50 backbone: 0.2s / 100 steps, R100 backbone: 0.25s / 100 steps), which is reasonable. Therefore, the addition of GUM does not bring too much computational burden and does not significantly increase the training time.

Table 10: Detailed Training Time of FR models.

| Training Data | Method | Average Training Time / 100 steps | Average Training Time / Epoch | IJB-C (1e-4) |
|---|---|---|---|---|
| MS1MV2 | R50, ArcFace | 15.33s | 1743.32s | 92.52 |
| | R50, ArcFace + PTSA | 17.57s | 1999.19s | 96.25 |
| | R50, ArcFace + PTSA + SDE (**R50, TopoFR**) | 17.77s | 2020.80s | **96.49** |
| MS1MV2 | R100, ArcFace | 20.16s | 2292.59s | 95.74 |
| | R100, ArcFace + PTSA | 23.10s | 2626.93s | 96.68 |
| | R100, ArcFace + PTSA + SDE (**R100, TopoFR**) | 23.35s | 2655.36s | **96.95** |

**6) Parameter Sensitivity:** To demonstrate the effect of the hyper-parameters $\alpha$ (*i.e.*, the balance coefficient of the ISA loss $\mathcal{L}_{sa}$) and $\xi$ (*i.e.*, the probability of RSP mechanism), we conduct additional experiments by setting different values of $\alpha$ and $\xi$, respectively. We use MS1MV2 dataset to train the R50 TopoFR and R100 TopoFR models, and evaluate their performance on the IJB-C benchmark. The results on "TAR@FAR=1e-4" are shown in Figures 6a and 6b.

We can observe that as $\alpha$ increases, the accuracy first rises and then falls. This is because an overly large $\alpha$ will cause the model to focus more on the structure alignment and less on the classification learning of the samples during training. Additionally, setting the parameter $\xi$ to a high value may introduce more unidentifiable face images and severely disturb the topological structure of the latent space, making it difficult for the ISA loss function $\mathcal{L}_{sa}$ to converge.

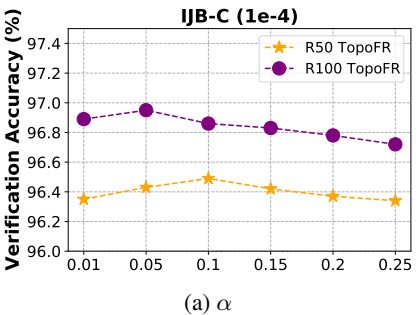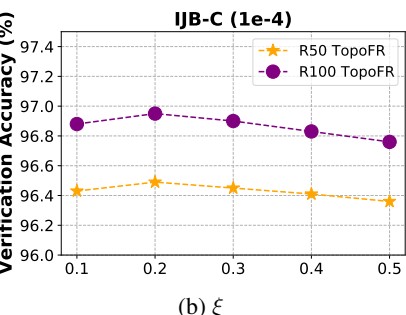

Figure 6: Parameter sensitivity analysis. (a) The effect of the hyper-parameter $\alpha$. (b) The effect of the hyper-parameter $\xi$.

**7) Comparison of Structure Discrepancy:** To further demonstrate the effectiveness of our PTSA strategy in preserving structure information, we utilize the **Bottleneck distance metric** [60, 18] to investigate the topological structure discrepancy between the input space and the latent space of ArcFace and TopoFR on IJB-C benchmark, as illustrated in Figure 7.

We can observe that our TopoFR model significantly reduces the structure discrepancy (*i.e.*, measured by the Bottleneck distance metric) between two spaces compared to the vanilla ArcFace model, and effectively preserves the structure information hidden in the large-sclae dataset.

**8) Visualization of Hard Samples:** We conduct a visual comparison experiment to visualize some hard samples that can be correctly classified by our method but cannot be correctly classified by existing method.

The visualization results illustrated in Figure 8 reflect the following observations: **(1)** Hard samples are usually blurry, low-contrast, occluded, or in unusual poses, so they are easily misclassified by existing method such as ArcFace model. **(2)** The ArcFace model assigns equal weight (*i.e.*, 1) to each sample. While our TopoFR model utilizes SDE strategy to adaptively assign weight (*i.e.*, SDS $\omega$ ) to each sample based on its prediction uncertainty $\omega_1$ and prediction accuracy $\omega_2$. Specially, SDE will assign higher SDS $\omega$ to hard samples, which can encourage the model to extract robust face features from these challenging samples, thereby effectively improving the model's generalization performance.

**9) Is the topological structure constructed in the input space robust enough? i)** In the input space, we first flatten the face images into vector features and then calculate their pair-wise distance matrix in order to construct Vietoris-Rips complex. And the dimension of face features in the pixel space is significantly higher than that of the features in the latent layer space. Notably, the expected $k$-th

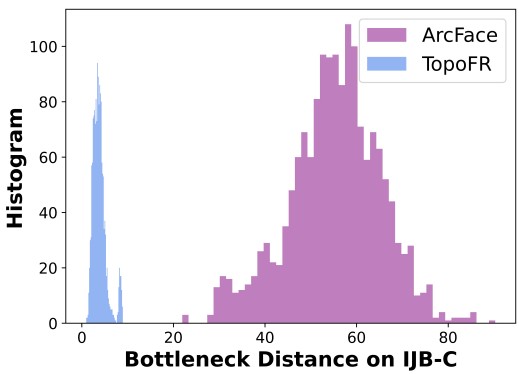

Figure 7: The topological structure discrepancy (*i.e.*, measured by the **Bottleneck distance metric**) of R50 TopoFR and R50 ArcFace on the IJB-C benchmark.

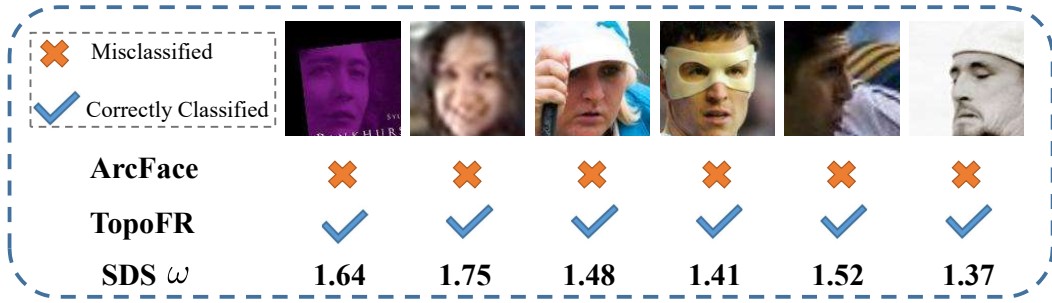

Figure 8: Visualization of hard samples.

Betti number $\mathbb{E}\left[\beta_k^{VR}(r)\right] = c_k n (nr^d)^{2k+1}$ ( $n$: data size, $d$: data dimension, $c_k$: constant) [97, 98] in topology theory also demonstrates that as long as the dimension of the data is sufficiently high, its underlying topological structure can be well constructed. Therefore, this can effectively capture the topological structure information hidden in the large-scale face datasets. **ii)** More importantly, during training, our RSP mechanism also effectively simulates the influence of multiple factors, such as lighting, occlusion, blur, etc., on the training samples, making the constructed topological structure robust enough to noise.

Overall, based on the analysis above, we believe that the topological structure constructed in the input space is sufficiently robust and can effectively guide the learning of the latent space structure.

### A.4 Limitation

Our model might has the following two limitations:

**(i)** Due to the introduction of the structure alignment strategy PTSA and hard sample mining strategy SDE, our TopoFR model requires longer training time (1.16x). For detailed training time analysis, please refer to the Table 10 in our appendix.

**(ii)** This work aims to improve the generalization performance of FR models by leveraging the inherent structure information in large-scale face training data. However, large-scale face training datasets in real-world scenarios inevitably contain a small amount of noisy labels. Our loss function does not assign any special treatment to these mislabeled samples. Since our hard sample mining strategy SDE assigns larger weights to hard samples that are misclassified or have high prediction uncertainty, mislabeled images may be wrongly emphasized. We believe that future works will be able to simultaneously address the challenges of structure alignment and label noise.

