# OpenReview forum: "TopoFR: A Closer Look at Topology Alignment on Face Recognition"
_NeurIPS.cc/2024/Conference — NeurIPS 2024 poster_

### Official Review · Reviewer_GA4X · 2024-07-11

**Soundness:** 4
**Presentation:** 3
**Contribution:** 3
**Rating:** 7
**Confidence:** 4

**Summary:**

This paper proposes a novel framework named TopoFR for face recognition (FR). The authors first discover that existing FR models struggle to effectively preserve the structure information hidden in FR dataset, and provide three specific observations:

(i) the topological structure in the input pixel space becomes increasingly complex as the amount of data increases;

(ii) the structure difference between pixel space and latent feature space increases as the amount of data increases;

(iii) the structure difference between two spaces decreases as the network architecture becomes deeper.

Considering that FR training datasets are typically massive and contain rich structure information, this paper aims to leverage these intrinsic structure information to enhance FR model's generalization performance. However, the authors notice that directly aligning the structures between latent space and input pixel space will result in inferior performance which might be overfitting. To solve this issue, TopoFR introduces a PTSA strategy to encode the structure information of input pixel space into latent space. Specifically, PTSA utilizes RSP to perturb the latent space structure and employs ISA to match the topological structures of two spaces, effectively mitigating structure discrepancy. Moreover, a novel hard sample mining strategy SDE is proposed to identify hard samples and minimize their impact on the latent space structure. Extensive experiments on various FR benchmarks demonstrate that TopoFR outperforms state-of-the-art methods. In addition, detailed ablation studies and analysis are conducted to provide a comprehensive understanding of the proposed method.

**Strengths:**

1) This paper is clearly written and well-motivated. The overall readability is high. Despite the significant progress made by existing FR models, these models rely on explicitly constraining inter-class and intra-class relationships. Unlike these existing works, this paper seeks to leverage the structure information in datasets to improve the FR model's generalization performance, which is a highly innovative attempt and can provide valuable insights for future research. Furthermore, it is novel to address the input-latent space structure degradation problem.

2) It is a very interesting idea to employ the GUM statistical distribution in designing the hard sample mining strategy SDE from the outliers detection perspective. The structure damage score SDS is similar to the area of noise labels, which is not well investigated in FR.

3) Experiments are good and the various benchmark results have produced state-of-the-art performance. Moreover, the model's ranking in the MFR competition shows its strong generalization performance.

**Weaknesses:**

There are some concerns that I would like the authors to address in order to enhance the overall comprehensiveness of the paper.

1) It is not clear if this method can be applied to recent proposed loss, e.g., AdaFace [1]. I recommend adding experiments to further validate the universality of the proposed method.

2) [Minor Comments]: Although the authors have discussed the model’s training and inference efficiency in Appendix, I suggest including comparisons of the model's GFLOPs. This would provide a more intuitive representation of the model’s computational cost.

Ref[1] Kim M, Jain A K, Liu X. Adaface: Quality adaptive margin for face recognition[C]//Proceedings of the IEEE/CVF conference on computer vision and pattern recognition. 2022: 18750-18759.

**Questions:**

1) EM algorithm is sensitive to the initialization of its parameters values. In Eq.(7), how to determine the initial values of parameters $\pi$, $\Sigma$, and $\Omega$ in the GUM statistical distribution?

2) In Eq.(2), does the ISA function loss $\mathcal{L}_{sa}$ guide the optimization of the entire network or only specific components of the network?

3) What distance criterion is used to construct the topological structure space of the data in Vietoris-Rips complex during model forward propagation?

4) In addition, in topological data analysis, different dimensions of topological holes analyze the underlying data structure information at different levels. In ISA loss, what dimension of topological holes does the persistent homology (PH) capture and analyze?

5) I am very curious about the reasons why the structure differences between the input pixel and latent spaces decrease as the network becomes deeper.

**Limitations:**

It's not discussed whether this method can be applied in other related FR methods, e.g. AdaFace. I suggest to further discuss this to increase the overall quality of this paper.

---

> ### Author Rebuttal · Authors · 2024-08-05
>
> We thank the Reviewer GA4X for the careful reading of the manuscript and the related comments, which are helpful to improve our paper.
> Our detailed point-by-point responses are provided below.
>
> **W1: It is not clear if this method can be applied to recent proposed loss AdaFace.**
> **A1:** According to your suggestion, we conduct additional experiments on AdaFace model to further validate the universality of our method, as shown in the following Table R1. \
> In comparison to the baseline model ArcFace, AdaFace is a highly advanced FR model. Consequently, integrating our method with AdaFace can notably enhance the model's verification accuracy across multiple benchmarks.
> These improved results validate the universality of our method.
>
> **Table R1: Verification performance (%) on IJB-C, IJB-B, CPLFW and CALFW.**
> | Training Data | Method | IJB-C(1e-4) | IJB-B(1e-4) | CPLFW | CALFW |
> | ------ | ------ | ------ | ------ | ------ | ------ |
> |  | R50 AdaFace | 96.27 | 94.42 | 92.83 | 96.07 |
> | MS1MV2 | R50 AdaFace + PTSA | 96.51 | 95.25 | 93.36 | 96.28 |
> |  | R50 AdaFace + PTSA + SDE | **96.64** | **95.38** | **93.50** | **96.39** |
>
> **W2: Includes comparison of the model's GFLOPs.**
> **A2:** Based on your valuable advice, we provide a comparison of model's GFLOPs, as shown in the following Table R2.
> As can be seen, our TopoFR has the same GFLOPs as the vanilla ArcFace model (Baseline) and some existing popular FR models (e.g., MagFace, AdaFace and CurricularFace), since we adopt the same network architecture.
>
> **Table R2: Comparison of Model's GFLOPs.**
> | Method | GFLOPs |
> | ------ | ------ |
> | R50 ArcFace | 6.3 |
> | R50 MagFace | 6.3 |
> | R50 AdaFace | 6.3 |
> | **R50 TopoFR** | 6.3 |
> | R100 ArcFace | 12.1 |
> | R100 MagFace | 12.1 |
> | R100 AdaFace | 12.1 |
> | R100 CurricularFace | 12.1 |
> | **R100 TopoFR** | 12.1 |
> | R200 ArcFace | 23.4 |
> | R200 AdaFace | 23.4 |
> | **R200 TopoFR** | 23.4 |
>
>
> **Q1: In Eq.(7), how to determine the initial values of parameters $\phi$, $\Sigma$, and $\Omega$ in the GUM statistical distribution ?**
> **A3:** We refer to prior work [S4] to set the initial values for the EM algorithm used in our GUM parameter estimation process.
>
> Ref.[S4] Using gaussian-uniform mixture models for robust time-interval measurement. TIM 2015.
>
>
> **Q2: In Eq.(2), does the ISA function loss $\mathcal{L}_{sa}$ guide the optimization of the entire network or only specific components of the network ?**
> **A4:** In fact, the ISA loss $\mathcal{L}_{sa}$ is only responsible for optimizing the parameters of the feature extractor $\mathcal{F}$.
>
> As stated in Section 4.1 (lines 140-146), since computing the pairwise distance matrices $\mathcal{M}^{\mathcal{X}}$ and $ \mathcal{M}^{\mathcal{\widetilde{Z}}}$ for Vietoris-Rips (VR) complexes $ \mathcal{V}\_\{\rho}(\mathcal{X})$ and $ \mathcal{V}\_\{\rho}(\mathcal{\widetilde{Z}})$ is a differentiable process, the ISA loss $\mathcal{L}_{sa}$ will perform gradient back-propagation through the two distance matrices to optimize the parameters of the feature extractor $\mathcal{F}$.
>
> **Q3: What distance criterion is used to construct the topological structure space of the data in Vietoris-Rips complex during model forward propagation ?**
> **A5:** As stated in Section 3 (lines 95-104), we utilize the Euclidean distance as the distance criterion to construct the Vietoris-Rips (VR) complex $\mathcal{V}_{\rho}$ and analyze the topological structure of the underlying space.
>
> **Q4: In ISA loss, what dimension of topological holes does the persistent homology (PH) capture and analyze ?**
> **A6:** As described in Section A.1 of the Appendix (lines 547-550), we preserve and analyze 0-dimension topological holes $H_{0}$ (e.g., connected components) in our ISA loss $\mathcal{L}_{sa}$.
> Because some preliminary experiments have shown that using the 1-dimension or higher-dimension topological holes only increases model’s training time without bringing clear performance gains.
>
>
> **Q5: I am very curious about the reasons why the structure differences between the input pixel and latent spaces decrease as the network becomes deeper.**
> **A7:** This is a very interesting question. When we obtained this finding in the experiment, we also contemplated its underlying reasons. \
> In our opinion, the deep neural network is actually a complex function mapping process, and each network layer can be regarded as a sub-function in this composite function. If the neural network is deeper, the mapping of each network layer will become smoother, and the data will lose less information during the mapping process. On the contrary, if the neural network is shallower, the mapping of each network layer will become sharper, and some intrinsic information of the data, such as structure information, is easily destroyed during the mapping process. Similar conclusions can also be found in the flow-based models [S5].
>
> Ref. [S5] Variational inference with normalizing flows. ICML 2015.

---

> > ### Comment · Reviewer_GA4X · 2024-08-09
> >
> > Thanks to the authors for the comprehensive response to my concerns.
> >
> > I have carefully read all the reviews and responses. The authors have addressed all of my concerns with clear explanations and additional experiment results, especially about the experiment of AdaFace + PTSA + SDE, making the benefits and novelty of the proposed method more convincing.
> >
> > I have raised the rating to reflect the comments.

---

> ### Author Response · Authors · 2024-08-10
> **Response to Reviewer GA4X**
>
> Dear Reviewer GA4X:
>
> Thank you for the positive feedback. We appreciate your efforts in reviewing our work. We will reflect your suggestions in the revised version to enable it to be a high-quality paper.
>
> Best Regards, Authors of 505.

---

### Official Review · Reviewer_68Et · 2024-07-12

**Soundness:** 3
**Presentation:** 3
**Contribution:** 2
**Rating:** 6
**Confidence:** 4

**Summary:**

This paper uses the topological structure alignment in face recognition tasks, and it proposes a Perturbation-guided Topological Structure Alignment (PTSA) strategy to align the topology of input image space and latent space. In this paper, Persistent Homology(PH) is used to verify that the complexity of input space topology will increase rapidly when there are many face images. In addition, this paper also puts forward some ideas and implementation of data mining.

**Strengths:**

+ The paper adopts topological structure alignment in face recognition tasks.
+ A new hard sample mining strategy SDE is proposed to reduce the adverse effects of hard samples on potential spatial structure.
+ Experiments are sufficient, and the experimental results prove that the model has excellent performance.

**Weaknesses:**

- Formula 7 may require a more specific explanation.
- The meaning of H0 in Figure 1, along with the distribution style and other differences, is best explained.
- The hard sample mining strategy SDE in this paper seems to have little correlation with Perturbation-guided Topological Structure Alignment.

**Questions:**

- Is this topological alignment strategy only for face recognition tasks?
- Is the process of alignment computationally intensive?

**Limitations:**

Yes

---

> ### Author Rebuttal · Authors · 2024-08-05
>
> We appreciate Reviewer 68Et for the thorough review of the paper and valuable comments that will aid in enhancing our paper.
>
> **W1: Formula 7 may require a more specific explanation.**
> **A1:** Based on your valuable suggestion, we provide an explanation of how to estimate the parameter set $\varphi = \\left \\{ \pi, \Sigma, \Omega \\right \\}$ of the Gaussian-uniform mixture (GUM) model using EM algorithm.
>
> First, we transform the original GUM model (Eq.(3)) into a more easily solvable statistical distribution (Eq.(6)) using the Bernoulli distribution-based sampling method. In this way, the maximum likelihood model of Eq.(6) is formulated as: $\max\limits\_\{\pi,\Sigma,\Omega} \displaystyle\prod_{i=1}^{n}p\left(\widehat{E}(\widetilde{g}\_\{i})|\widetilde{x}\_\{i}\right )$.
> Then, we can use the EM algorithm to solve the maximum likelihood problem in order to estimate the parameter set $\varphi$ of GUM. The specific iterative updating formulas of EM algorithm are shown in Eq.(7).
>
> Specially, at each iteration, EM alternates between evaluating the expected log-likelihood (E-step) and updating the parameter set $\varphi$ (M-step). In Eq.(7), the **E-step** aims to evaluate the posterior probability $h_{\varphi}^{(t+1)}$ of an sample $\widetilde{x}\_\{i}$ to be hard sample using the iterative formula $h\_\{\varphi}^{(t+1)}(\widetilde{x}\_\{i})$, where $(t+1)$ denotes the EM iteration index. The **M-step** updates the parameter set $\varphi$ using the iterative formulas $\pi^{(t+1)}$, $\Sigma^{(t+1)}$ and $\Omega^{(t+1)}$, where $\eta_{1}$ and $\eta_{1}$ are the first-order and second-order centered data moments, respectively.
>
> **W2: The meaning of H0 in Figure 1, along with the distribution style and other differences, is best explained.**
> **A2:** As described in Sec.3 (lines 105-109), homology is an algebraic structure that analyzes the topological features of a simplicial complex in different dimension $j$, including connected components ($H_{0}$), cycles ($H_{1}$), voids ($H_{2}$), and higher-dimensional topological features ($H_{j},j\geq 3$). By tracking the changes in topological features $H_{j}$ across different dimensions $j$ as the scale parameter $\rho$ increases, we can obtain the multi-scale topological information of the underlying space.
>
> Thus, $H_{0}$ is the $0$-th dimension homology in the persistence diagram, which captures the $0$-th dimension topological feature of the underlying space. Notably, low-dimension topological features (e.g., $H_{0}$) can roughly reflect the topological structure of the space, while high-dimension topological features (e.g., $H_{3}$, $H_{4}$) aim to capture the intricate details of the space's topological structure.
>
> Moreover, in topology theory, an increase in the number of high-dimensional topological features within a space corresponds to a more complex topological structure. As illustrated in Figs 1(a)-1(d), as the amount of face data increases, the persistence diagram contains more and more high-dimensional homology (e.g., $H_{3}$ and $H_{4}$), indicating that the input space contains an increasing number of high-dimensional topological features. Thus, this phenomenon shows that the topological structure of the input space is becoming more and more complex.
>
> **W3: The hard sample mining strategy SDE seems to have little correlation with Perturbation-guided Topological Structure Alignment (PTSA).**
> **A3:** As stated in Sec.1, the large-scale face dataset contains rich topological structure information. However, we find that these structure information is not effectively preserved in the latent space of existing FR models, as verified in Fig 2 (lines 31-43). Existing studies on FR have overlooked this issue, which limits the generalization of FR models. To solve this problem, we propose the PTSA strategy. Specially, our PTSA strategy directly extracts the topological structure information in the face dataset from the input pixel space and then encodes these powerful structure information into the latent space by aligning the topological structures of the input and latent spaces.
>
> However, as stated in Sec 4.2 (lines 170-177), we experimentally find that some low-quality samples (e.g., hard samples) can easily be encoded to abnormal positions in the latent space during training, which disrupts the latent space’s topological structure and further hinder the alignment of topological structures. Thus, we propose the SDE strategy to identify hard samples with significant structure damage and effectively mitigate their adverse impact on the latent space’s topological structure by guiding them back to their appropriate positions during optimization.
>
> Hence, our PTSA strategy and SDE strategy are complementary to each other. Additionally, the ablation study in Table 3 also validate the complementarity of two strategies.
>
> **Q1: Is this topological alignment strategy only for face recognition (FR) tasks?**
> **A4:** In addition to FR systems, the proposed PTSA strategy can also be applied to Image Retrieval and Large Language Model (LLM), since these tasks all require large-scale datasets for training.\
> Existing works on Image Retrieval and LLM seem to neglect the utilization of topological structure information hidden in large-scale datasets. Thus, we believe that applying the topological structure alignment technique to these tasks can effectively improve the representation power of features and the generalization ability of models.
>
> **Q2: Is the process of alignment computationally intensive?**
> **A5:** In fact, we have provided an analysis of the model training time in our manuscript (lines 637-650). Please refer to Table 13 in the Appendix. For example, compared to the R50 ArcFace (Baseline), our R50 ArcFace + PTSA requires about 2.24 seconds more training time per 100 steps. These results indicate that introducing PTSA strategy leads to significant performance gains with only a small increase in training time, which is acceptable.

---

> > ### Comment · Reviewer_68Et · 2024-08-13
> >
> > Thanks for your responses. I have changed my rating to Weak Accept.

---

> > > ### Author Response · Authors · 2024-08-13
> > > **Response to Reviewer 68Et**
> > >
> > > Dear Reviewer 68Et:
> > >
> > > We thank your response and appreciation of our work and rebuttal. We will make sure to incorporate the discussions into our revision to enable it to be a high-quality paper.
> > >
> > > Best Regards, Authors of 505.

---

### Official Review · Reviewer_SWuk · 2024-07-15

**Soundness:** 3
**Presentation:** 3
**Contribution:** 3
**Rating:** 6
**Confidence:** 4

**Summary:**

The paper introduces a new method to improve the topological structures of facial features in the latent space. By exploring the topological structure alignment in face recognition, the authors propose a new structural alignment strategy PTSA to align the structures of origin input space and feature space. The experimental results on various benchmarks have verified the effectiveness of the proposed method.

**Strengths:**

- Based on the persistent homology method, the paper provides a comprehensive analysis of the correlation between the number of data samples and topological structures in the latent space.
- The proposed modules in the method, Perturbation-guided Topological Structure Alignment and Structure Damage Estimation, are well motivated and seem to improve the topological structure of facial features.
- The authors achieve competitive results on the standard benchmarks, especially ranking #2 on the MFR-Ongoing leaderboard.

**Weaknesses:**

- While I acknowledge the competitive results of the proposed approach, the performance improvement of the method in several benchmarks remain minor. For example, the results of the method on IJB-B and IJB-C benchmarks are minor compared to prior methods on the same backbone.

- The technical approach seems to be an incremental one. In particular, the authors adopt the ArcFace framework and introduce two additional modules on top of the ArcFace method, including Perturbation-guided Topological Structure Alignment and Structure Damage Estimation. The first module is basically a metric to compare the feature distribution between feature space and original input space, similar to the Gromov-Wasserstein distance. However, the ArcFace loss itself is also an efficient approach to learning the facial feature representation in the latent space. I am not sure whether the Perturbation-guided Topological Structure Alignment significantly improves the structure information or not.

- Why the authors claim the proposed approach improve the topological structure of the feature space, I could not see this point has been verified via either experimental results or visualization. It would be better if the authors visualize the feature distributions of faces on the latent space (via PCA or T-SNE).


- The author claims that "to the best of our knowledge, how to effectively mine the potential structure information in large-scale face data has not investigated". I do not agree with this claim since the other methods, e.g., ArcFace, SphereFace, CoseFace, etc, are also considered as approaches to improve the structures/distributions of facial features in the deep latent space via the well design loss functions. I think the statements made by the authors are over-claimed.

**Questions:**

Please refer to my weakness section. In addition, I have another question related to the Perturbation-guided Topological Structure Alignment. Are the persistence diagrams and persistence pairings computation in the ISA loss differentiable for backpropagation? Can the authors detail this computational process?

**Limitations:**

The authors have discussed their limitations in the appendix. However, it will be better if the authors discuss the broader impact of the proposed method.

---

> ### Author Rebuttal · Authors · 2024-08-06
>
> We sincerely appreciate Reviewer SWuk for the careful reading and the insightful comments, which are helpful in improving our paper. Our detailed point-by-point responses are provided below.
>
> **W1: The results of the method on IJB-B and IJB-C are minor.**
> **A1:** Due to the characters limit, we will respond to you in the **global rebuttal area**.
>
> **W2: The technical approach seems to be an incremental one. The PTSA strategy is similar to the Gromov-Wasserstein distance.**
> **A2:** Due to the characters limit, we will respond to you in the **global rebuttal area**.
>
> **W3: I am not sure whether the PTSA strategy significantly improves the structure information or not. Why the authors claim the proposed approach improve the topological structure of the feature space? It would be better to visualize the feature distributions.**
> **A3:** As shown in Figures 2(a) and 2(b), we experimentally find that there are significant topological structure discrepancy between the input space and the latent space of existing FR models. This observation indicates that the topological structure information of the large-scale dataset is not effectively preserved in the latent space and may even be destroyed. Therefore, we propose the PTSA strategy to align the topological structure of two spaces. The visualization results in Figs 2(d), 5 and 7 have validated that with the help of our PTSA strategy, the topological structure discrepancy between two spaces have significantly decreased. In fact, the results in Figs 2(d), 5 and 7 can quantify topological structure discrepancy more objectively and accurately than t-SNE. Moreover, the ablation study results in Tab 8 of Appendix have also validated that the addition of PTSA can implicitly enhance the intra-class compactness and inter-class separability of facial features.
>
> **t-SNE:** Based on your insightful advice, we sample 10 identities face images from MS1MV2 dataset and utilize the t-SNE to visualize the facial features learned by ArcFace and ArcFace + PTSA. The results are shown in the Figure 1 of the **global response PDF file**. We can observe that using the proposed PTSA strategy to reduce the topological structure discrepancy between two spaces can greatly enhance the discriminative power of the learned facial features (e.g., with better inter-class separability and intra-class compactness), thereby boosting the FR model's generalization performance. The t-SNE results are consistent with our aforementioned experimental results.
>
> **W4: Other methods are also considered as approaches to improve facial features distributions in latent space via well design loss functions.**
> **A4:** Existing FR methods, such as ArcFace, CosFace, SphereFace and CurricularFace, tended to design some margin-based softmax loss functions to explicitly encourage facial features with better inter-class separability and intra-class compactness in feature space. However, the experimental results in Figs 2(a) and 2(b) have shown that the latent space's topological structure of existing models may be destroyed, which limits FR model's generalization. Thus, unlike these existing FR works that directly force features to become more discriminative in the feature space, our PTSA strategy aims to leverage the intrinsic topological structure information in training dataset (i.e.,from the input pixel space) to guide the construction of the latent space's topological structure and the learning of facial features.  Detailed experiments have indicated that our method can effectively reduce topological structure discrepancy and boost model's generalization.
>
> In general, leveraging the topological structure information in dataset to enhance the model's generalization is a different research direction from designing the margin-based softmax loss functions to improve facial features distribution in the deep feature space.
>
> **Q1: Are the persistence diagrams (PDs) and persistence pairings (PPs) computation in ISA loss differentiable for backpropagation? Detail this computational process.**
> **A5:** Yes, the computation of PDs and PPs in ISA loss is differentiable for back-propagation.
>
> **1) Detailed computation process:** Persistent homology (PH) is a method used to analyze the topological structure information of complex point clouds (e.g., mini-batch data). In the original input space, we first flatten the mini-batch face images into vector features to model point clouds $\mathcal{X}$, and then calculate their pairwise distance matrix to construct the Vietoris-Rips (VR) complex $\mathcal{V}\_\{\rho}(\mathcal{X})$. Similarly, we can construct another VR complex $\mathcal{V}\_\{\rho}(\widetilde{\mathcal{Z}})$ for the perturbed latent features $\widetilde{\mathcal{Z}}$.
> After that, we use the Ripser package [S3] to implement the PH to extract topological features from two VR complexes, and obtain their corresponding PDs and PPs. The discrepancy between the input space's PD and latent space's PD represents the topological structure discrepancy of two spaces. However, directly calculating the distance between two PDs is computationally expensive. We thus turn to utilize PPs to estimate the discrepancy between two PDs.
> PPs contain indices of simplices that are related to the birth and death of the topological feature. In this case, we can calculate the discrepancy between two PDs by utilizing PPs to retrieve the differences between two pairwise distance matrices.
>
> Ref.[S3] Ripser: Efficient Computation of Vietoris–Rips Persistence Barcodes. 2021.
>
> **2) Optimization of model params w.r.t $\mathcal{L}_{sa}$:** Please refer to the 6th answer **A4** to **Reviewer GA4X**.
>
>
>
> **Q2: Discuss the broader impact.**
> **A6:** Due to the characters limit, we will respond to you in the **global rebuttal area**.

---

> > ### Comment · Reviewer_SWuk · 2024-08-12
> > **Feedback to Author Rebuttal**
> >
> > Thank you very much for your rebuttal. It has addressed my concerns.  I hope you can update your answers in the revised paper. It will help to improve the paper quality. I decided to increase my score to 6.

---

> > > ### Author Response · Authors · 2024-08-12
> > > **Response to Reviewer SWuk**
> > >
> > > Dear Reviewer SWuk:
> > >
> > > Thank you for your recognition of our work. We appreciate your time and efforts in reviewing our work. We will incorporate your valuable suggestions into the revised version to ensure that it becomes a high-quality paper.
> > >
> > > Best Regards, Authors of 505.

---

### Author Rebuttal · Authors · 2024-08-06

Dear **Reviewer SWuk**, here is our response to some of the concerns you raised.

**W1: The results of the method on IJB-B and IJB-C are minor.**
**A1:** (1) Notably, when using a shallow backbone such as ResNet-50, our method showcases remarkable performance gains on IJB-B, IJB-C and other benchmarks, as shown in Tabs 2 and 4, compared to the leading competitors. This is because the shallow backbone has larger topological structure discrepancy between the input and latent spaces (as verified in Figure 2(b)), and our method can effectively mitigate the structure discrepancy and boost the model's generalization performance. Furthermore, when using extremely large-scale datasets such as Glint360K and WebFace42M, our method can achieve more significant accuracy improvements, as shown in Tabs 1 and 2. Since larger-scale dataset contains richer topological structure information, which more clearly indicates the effectiveness of our method in using topological structure information to boost model's generalization.

(2) On some low-quality (e.g., contains many blurry face images) and challenging FR benchmarks, such as MFR-Ongoing, MegaFace, MegaFace-Refined, TinyFace, CPLFW and CALFW, our method can achieve clear performance gains compared to the competitors, as shown in Tabs 4, 5 and 6 of the Appendix. Additionally, as shown in Tab 7 of Appendix, our method can obtain a significant improvement in performance when combined with the lightweight face recognition (FR) network backbone MobileFaceNet.

(3) The accuracy of existing FR methods on the IJB-B and IJB-C benchmarks is nearing saturation, as these two datasets contain many high-quality face images. Our proposed method is still able to achieve performance gains, surpassing the ResNet-based SOTA competitor AdaFace [S1] and the ViT-based SOTA competitor TransFace [S2]. Notably, in FR task, a performance improvement of approximately 0.2% to 0.3% on the IJB-C (1e-4) and IJB-B (1e-4) benchmarks is quite substantial. While our R200 TopoFR trained with Glint360K achieves maximum performance gains of 0.49% on IJB-C (1e-4) and 0.47% on IJB-B (1e-4) compared to the SOTA method AdaFace, respectively. In general, our method achieves SOTA performance on various benchmarks using different backbones, implying its strong generalization ability.

Ref.[S1] Adaface: Quality adaptive margin for face recognition. CVPR 2022.\
Ref.[S2] Transface: Calibrating transformer training for face recognition from a data-centric perspective. ICCV 2023.

**W2: The technical approach seems to be an incremental one. The PTSA strategy is similar to the Gromov-Wasserstein (GW) distance.**
**A2:** As mentioned in Sec.1 (lines 31-43), we are the first to discover that existing FR models struggle to effectively preserve the topological structure of face data in the latent space, and we provide some novel observations (as depicted in Figs 1 and 2).
However, existing FR studies have overlooked this issue, which severely limits the models' generalization. Based on this motivation, we are the first to propose aligning the topological structures of two spaces to improve the model's generalization performance. A series of experimental results have also shown that the topological structure information in the dataset is quite crucial for FR task. Thus, our method is not an incremental work, but a novel exploration.

Moreover, the Gromov-Wasserstein (GW) distance metric is different from the topological structure distance metric. Specially, the topological structure models the relationships and connectivity between data by analyzing the evolution progress of the simplicial complex constructed from the data in high-dimensional metric space. While the GW distance is often used to align the mainfold structures of two feature distributions. Mainfold structure primarily models the similarity between data based on the similarity matrix, which is a different concept from topological structure. Thus, our PTSA strategy does not simply match feature distributions, but rather aligns the topological structures of the input and latent spaces in the high-dimensional metric space.


**Q2: It will be better if the authors discuss the broader impact of the proposed method.**
**A6:** Thanks for your valuable advice.
It would be good to mention that the utilization of face images do not have any privacy concern given the datasets have proper license and users consent to distribute biometric data for research purpose. We address the well-defined face recognition task and conduct experiments on publicly available face datasets. Therefore, the propose method does not involve sensitive attributes and we do not notice any negative societal issues.

**t-SNE Visualization**: The t-SNE visualization results mentioned in A3 are shown in the Figure 1 of the global response PDF file.

**To all Reviewers:** If you have additional concerns, please let us know and we will do our best to address them. We appreciate your time and efforts in reviewing our work.

---

### Decision · Program_Chairs · 2024-09-25

**Decision:**

Accept (poster)

**Comment:**

The rebuttal provided clarifications about the proposed method and its analysis that were useful for assessing the paper's contribution and responded adequately to most reviewer concerns. All reviewers recommend acceptance after discussion (with two weak accepts and one accept), and the ACs concur. The final version should include all reviewer comments, suggestions, and additional clarifications from the rebuttal.